# In Vitro Antioxidant, Antibacterial and Mechanisms of Action of Ethanolic Extracts of Five Tunisian Plants against Bacteria

Khaoula Nefzi [1], Mariem Ben Jemaa [2] , Mokhtar Baraket [1] , Sarra Dakhlaoui [2] , Kamel Msaada [2,*] and Zouheir Nasr [1]

[1] LR11INRGREF0 Laboratory of Management and Valorization of Forest Resources, National Research Institute of Rural Engineering, Water and Forests (INRGREF), Carthage University, Ariana 2080, Tunisia; nefzikhaoula@hotmail.com (K.N.); moktar.baraket@gmail.com (M.B.); zouheirnasr84@gmail.com (Z.N.)

[2] Laboratory of Aromatic and Medicinal Plants, Biotechnology Center in Borj-CedriaTechnopark, P.O. Box 901, Hammam-Lif 2050, Tunisia; mariembenjemaa@yahoo.fr (M.B.J.); saradakhlaoui@gmail.com (S.D.)

* Correspondence: msaada.kamel@gmail.com; Tel.: +216-22205878

**Abstract:** *Pistacia lentiscus*, *Rosmarinus officinalis*, *Erica multiflora*, *Calicotome villosa*, and *Phillyrea latifolia* were considered important medicinal herbs and were used to treat various ailments. The present study was designed to evaluate the antioxidant and antimicrobial activities of ethanolic extracts (EEs). *P. lentiscus* and *R. officinalis* were the richest species in phenolic compounds. Similarly, both species showed the highest values of flavonoids. While the EEs of *P. lentiscus*, *E. multiflora*, and *C. villosa* had higher amounts of tannins. These phenolic compounds were evaluated by two different tests, namely diphenyl picrylhydrazyl (DPPH) and ferric iron-reducing power (FRAP). The $IC_{50}$ values were found to be significant ($p < 0.05$) for *P. lentiscus* and *E. multiflora*. Similarly, both plants showed the highest ferric-reducing antioxidant power (FRAP). This study has been conducted to evaluate the antibacterial potential of EEs against selected bacteria—Gram-positive bacteria (*Staphylococcus aureus* ATCC 29213, *Listeria monocytogenes* ATCC 7644) and Gram-negative bacteria (*Escherichia coli* ATCC 8739, *Salmonella typhimurium* NCTC 6017)—and determine their modes of action. The ethanolic extracts inhibited bacterial growth by producing concentration-dependent zones of inhibition. Treatment with these extracts at their minimum inhibitory concentrations (MICs) showed a significant reduction ($p < 0.05$) in the viability of bacteria. The extracts did not induce total lysis. Bacteria organisms treated with EEs at MICs showed a significant ($p < 0.05$) loss of tolerance to NaCl (5%). Our results highlighted the use of plant extracts as natural antibacterials that can be safely used in health care and led to the understanding of the antibacterial mechanism of plant extracts.

**Keywords:** medicinal plants; ethanolic extracts; antioxidant activities; antibacterial activity; antibacterial mechanism of actions

## 1. Introduction

Aromatic and medicinal plants (AMPs) constitute, by their diversity, a reservoir of biologically active substances. These plants represent between 20,000 species of plants used in the world for pharmaceutical, therapeutic, agri-food, cosmetological, and industrial purposes [1–4]. The Mediterranean region, one of the world's biodiversity hotspots, is known for its high richness of vegetation, especially aromatic and medicinal plants [5,6]. While in Tunisia, the diversity of the flora is characterized by a richness of species (estimated at 2100 species) in addition to 149 medicinal species and 38 aromatic plants [2,3]. Aromatic and medicinal plants contain various phytochemicals, such as tannins, alkaloids, terpenoids, and flavonoids, which are characterized by antioxidant and antimicrobial powers [7].

Regarding the studied species, they are characterized by their antioxidant, antimicrobial, anti-inflammatory, and anti-cancer power. *Calicotome villosa* (belonging to the Fabaceae family) and *Erica multiflora* (belonging to the Ericaceae family) are species used as

antitumor remedies, and they are also used to treat boils, skin abscesses and chilblains [8,9]. *Phillyrea latifolia* (belonging to the Oleaceae family) and *Pistacia lentiscus* (belonging to the Anacardiaceae family) possess hypotensive action [10], and these species were studied for alterations in gastrointestinal nematode larvae [11]. *Rosmarinus officinalis* (belonging to the Lamiaceae family) has antioxidant, vasodilator, and anti-inflammatory properties [12].

In traditional medicine, crude extracts of different parts of aromatic and medicinal plants, including the fruit, flower, stem, root, and twigs, were largely used to treat certain bacterial diseases [13]. Recently, the use of plant-derived natural products as treatment strategies has gained prominence, and this is mainly due to the fight against the secondary effects of allopathic forms of medicine and the increasing incidence of antibiotic resistance [5,14]. Additionally, this undesirable change in antibiotic efficacy has prompted scientists to find another approach that can play a similar role with greater benefits and fewer disadvantages [15]. As a result, AMP is considered one of the "classic" forms of health care. Medicinal plants possess an important role in the treatment of complementary strategies and alternative forms of medicine because they produce a large variety of natural compounds with high therapeutic properties [16,17].

Several plant species have been tested for their antimicrobial activity, but the vast majority of these plants have not been adequately investigated [18]. Furthermore, information on the mechanism of action of antibacterial activity of the AMP extracts and associated bioactive compounds are still lacking [19]. It is well known that the general mechanism of antimicrobial compounds implicates the disruption of membrane potential or the permeabilization of the bacterial membrane or inhibition of macromolecular production [17]. Moreover, classical antibiotics act on microbes by affecting the synthesis of proteins and nucleic acids, the cell wall, and metabolite activity. Therefore, it would be beneficial if the mechanism of action of the supplemental antimicrobial agent was not the same as that of the antibiotic employed. AMP extracts or active ingredients of plant-based products that are not toxic to cells would be the best choice to be combined with antibiotics [1–4,17,20,21].

Consequently, the aim of the current study is first to determine the secondary metabolites of *Pistacia lentiscus*, *Rosmarinus officinalis*, *Erica multiflora*, *Calicotome villosa*, and *Phillyrea latifolia* and to estimate their antioxidant activities. Then, the potential protective effects of these extracts were evaluated against certain pathogenic bacteria. Finally, the mechanisms of action of the antibacterial activity of studied extracts were studied against *Escherichia coli*, *Salmonella typhimurium*, *Listeria monocytogenes*, and *Staphylococcus aureus* strains.

## 2. Materials and Methods

### 2.1. Plant Material

Leaves of five Tunisian medicinal plants (*Pistacia lentiscus*, *Rosmarinus officinalis*, *Erica multiflora*, *Calicotome villosa*, and *Phillyrea latifolia*, i.e., three plants per species and plot) were collected in the spring (27th March) of 2021 from the region of Zaghouan (northeast of Tunisia: 36°22′ N, 10°07 E, 340 m above sea level), transported in a portable cooler and identified in our laboratory in the National Research Institute of Rural Engineering, Waters and Forests (INRGREF) and then stocked at −20 °C until freeze-dried. The leaves of the species were crushed in a 2 mm diameter grinder.

### 2.2. Polyphenol Extraction

#### 2.2.1. Preparation of Ethanolic Extracts (EE)

In 10 mL of pure ethanol, one gram of the plant's powder was suspended. The suspension was stirred with a magnetic stirrer and kept at 4 °C in the dark for 24 h. On ashless filter paper, the mixture was then filtered. Finally, the extract was placed at 4 °C in the dark for future utilization.

#### 2.2.2. Total Polyphenol Contents

Total polyphenols were assessed by Folin–Ciocalteu reagent in accordance with the Dewento method [22]. A total of 500 μL of distilled water was used to make the suspension

and 125 microliters of Folin–Ciocalteu reagent to 125 microliters of EEs. After allowing the liquid to sit for 3 min, sodium carbonate ($Na_2CO_3$, 7%) was added to adjust the final volume to 3 milliliters. The suspension was then incubated at room temperature for 90 min in the dark. A UV/VIS spectrophotometer was used to measure the absorbance at 760 nm. (Cecil Aurius Series CE 2021, SelectScience, Bath, UK). The standard range is prepared with gallic acid. Total polyphenol contents are expressed as milligrams of gallic acid equivalent per gram of dry weight (mg GA/gDW).

### 2.2.3. Total Flavonoid Contents

The flavonoid contents were assessed as described by Dewento et al. [22] using sodium nitrite ($NaNO_2$, 5%), aluminum chloride ($AlCl_3$ $6H_2O$, 10%) and sodium hydroxide (NaOH, 1 M). A UV/VIS spectrophotometer (Cecil Aurius Series CE 2021, SelectScience, Bath, UK) was used to measure the absorbance at 510 nm. Catechin was used to produce the standard range, which ranged from 50 to 500 mg/mL. The amount of flavonoids is measured in mg catechin equivalent per gram of dry weight (mg EC/g DW).

### 2.2.4. Condensed Tannin Content

Broadhurst and Jones' [23] vanillin assay was used to determine tannin in plant EEs. Three mL of vanillin (4%) and then 1.5 mL of diluted EEs was added to 1.5 mL of pure sulfuric acid ($H_2SO_4$). The mixture was incubated for 15 min at room temperature after homogenization. At 500 nm, the absorbance was compared to a blank containing pure acetone [24]. Condensed tannin contents are determined by using a standard curve (0 to 500 µg/mL catechin). As for flavonoids, amounts of condensed tannins were determined in mg of catechin equivalent per gram of DW.

### *2.3. Antioxidant Activity Evaluation*
### 2.3.1. Evaluation of the Total Antioxidant Activity (TAA)

The TAA of the studied plant extracts was estimated using the colorimetric assay at 695 nm using a visible–UV spectrophotometer (Cecil Aurius Series CE 2021) according to Prieto and Aguilar [25]. The TAA was expressed as mg GAE/g DW (milligrams of Gallic Acid equivalent per gram of dry weight).

### 2.3.2. DPPH Radical Scavenging Activity

Burits and Bucar [26] and Cuendet et al. [27] have used a spectrophotometric approach to measure the scavenging potential of the examined ethanolic extracts, which was based on the reduction a methanol solution of DPPH radical.

The inhibition capacity of the DPPH free radical (expressed in percent %) was estimated by the following equation:

$$IC\ (\%) = \frac{Ablank\ -\ Asample}{Ablank}100 \tag{1}$$

where Ablank and Asample are the control and sample absorbances at 517 nm, respectively. The graph showing inhibition percentage versus extract concentration yielded the extract concentration that provided 50% inhibition ($IC_{50}$).

### 2.3.3. Determination of the Ferric-Reducing Antioxidant Power (FRAP)

The method proposed by Oyaizu [28] was used to determine the reducing power. One mL of the extract at various concentrations (0.1 to 1.5 mg/mL) is mixed with 2.5 mL of phosphate buffer (0.2 mol/L, pH 6.6) and 2.5 mL of $K_3Fe(CN)_6$ (1%) in this technique. At 50 °C, the resultant mixture is incubated for 20 min. Following this incubation, 2.5 mL of TCA (10%) was added to stop the reaction, which was then centrifuged at $650 \times g$ for 10 min at room temperature. Finally, the supernatant is diluted with 2.5 mL distilled water and 0.5 mL $FeCl_3$ (0.1%) (2.5 mL). At 700 nm, the absorbance is measured (the blank is the extraction buffer). Ascorbic acid or BHA (0.01–1 mg/mL) is used as a positive control. The

data are given in effective concentration ($EC_{50}$, g/mL), which is the extract concentration that corresponds to an absorbance of 0.5.

*2.4. Antibacterial Activity Assessment*

The antibacterial efficacies of the ethanolic extract of *C. villosa*, *P. lentiscus*, *R. officinalis*, *E. multiflora*, and *P. latifolia* were tested against microorganisms, including Gram-positive bacteria (*Staphylococcus aureus* ATCC 29213; *Listeria monocytogenes* ATCC 7644) and Gram-negative bacteria (*Escherichia coli* ATCC 8739; *Salmonella typhimurium* NCTC 6017). Bacterial Species were generously provided by the Biotechnology Center in Borj Cédria Technopole, Tunisia.

The antibacterial efficiency of tested ethanolic extracts was qualitatively and quantitatively evaluated.

The disc diffusion method was employed to assess the antibacterial potential of the extracts that were evaluated [29]. Fifteen μL of the EEs (Concentrations = MIC) were added to the sterile filter paper discs (6 mm diameter). Then, these discs were placed on the agar that had been previously inoculated with the selected bacteria. Gentamicin (10 μg/disc) was used as a positive control. Negative control corresponded to a disc without a sample (disc contains ethanol to determine the solvent activity). The culture plates were incubated for 24 h at room temperature (37 °C). The qualitative antibacterial activity of the tested extracts was determined by measuring the diameter of the inhibited growth zone (with the diameter of the 6 mm disc). Three repetitions were carried out for these experiments.

Secondly, the determination of the minimum inhibitory concentrations (MIC) and the minimum bactericidal concentrations (MBC) of tested ethanolic extracts presented the quantitative antibacterial activity and were estimated using the broth dilution method as described by Cosentino et al. [30]. Different concentrations of ethanolic plant extracts, ranging from 1.56 to 50 mg/mL, were prepared using the serial dilution technique, including a growth control and a sterility control.

After incubation at 37 °C for 24 h, Petri plates were inoculated with 10 μL of each dilution and incubated overnight at 37 °C. The growth of tested isolates (MIC) was determined by the lowest number of colonies formed in each bacterium at 37 °C overnight. The MBC refers to the minimum concentration of the extract at which isolates were killed completely. Experiments were performed at least in triplicates.

*2.5. Mode of Action of Ethanolic Extract*

The mode of action of the antibacterial activity of *C. villosa*, *P. lentiscus*, *R. officinalis*, *E. multiflora*, and *P. latifolia* EEs was assessed against *Escherichia coli*, *Salmonella typhimurium*, *Listeria monocytogenes*, and *Staphylococcus aureus* was investigated using different tests.

2.5.1. Time Kill Assay

Time-kill research evaluates the reduction in bacterial count in the presence of extracts at their MIC over several hours to characterize the antibacterial activity of the tested extract. Activities of tested products used were assessed by the determination of the reduction in the number of colony-forming units (CFUs) per mL over 0, 2, 4, and 24 h, as reported by Carson [31]. By graphing viable colony counts (CFU/mL) against time, the killing rate was calculated. Experiments were performed in triplicates.

2.5.2. Bacteriolysis

To evaluate the possible bacteriolytic action of tested plant extracts, absorbance at 620 nm was determined: an unlysed bacterium absorbs at 620 nm; thus, bacteriolysis may occur if the absorbance at 620 nm decreases over time [31].

Two bacterial colonies, obtained from an 18 h culture, were used to inoculate 400 mL of nutrient medium and incubated at 37 °C for 18 h under agitation. The bacteria were then separated from the culture medium by centrifugation at 10,000 rpm for 12 min at 4 °C. Sodium phosphate buffer (PBS) was used to wash the bacterial pellet (wash 2 times). The

pellet was suspended in PBS-Tween 80 (0.01%, *v/v*). The bacterial culture was calibrated so that a 1/100 dilution's optical density at 620 nm (OD620) was 0.310. At a concentration corresponding to each MIC, the EEs were incorporated into the bacterial suspension. The control suspension was mixed with PBS-T. A stirring with a Vortex mixer is carried out for the suspensions obtained for the 20 s. Samples (10 μL) were taken in duplicate at 0, 30, 60, 120 min, and 24 h, serially 10-fold diluted, and the OD620wasmeasuredimmediately).

### 2.5.3. Loss of Salt Tolerance

The ability of *E. coli* and *S. aureus* strains treated with the tested EEs to grow on nutrient agar (NA) supplemented with NaCl was studied according to the method previously described by Carson et al. [31]. Untreated bacterial suspensions were treated for 30 min with the EEs of *C. villosa*, *P. lentiscus*, *R. officinalis*, *E. multiflora*, and *P. latifolia* at at a concentration corresponding to each MIC was plated on NA containing NaCl at 0 to 50 g/L. After incubation at 37 °C for 24 h, colonies were counted. The number of CFUs per milliliter obtained on the nutrient agar with or without NaCl was compared, and the result was expressed in terms of percentage.

### 2.6. Statistical Analyses

Statistical analysis was conducted using SAS 9.4 (Texas A&M University, College Station, TX, USA) by analysis of variance (ANOVA) in addition to Pearson's test for the evaluation of the significance of the difference between parameters. Data values were expressed as means $\pm$ standard deviation using Duncan multiple range tests at $p < 0.05$. For each measurement, at least three replicates were performed.

## 3. Results

### 3.1. Determination of Total Phenols, Flavonoids, and Tannin Contents

The total polyphenol, flavonoid, and tannin contents of the leaves revealed significant differences ($p < 0.05$) between ethanolic extracts (Figure 1). The results showed that *P. lentiscus* and *R. officinalis* were the richest species in phenolic compounds with values of 57.83 $\pm$ 3.58 and 64.16 $\pm$ 4.71 mg GAE/gDW, respectively (Figure 1A). In addition, the EEs of both species were also the richest in flavonoids compared to the other species studied. They had 8.36 $\pm$ 0.34 mg CE/gDW for *P. lentiscus* and 9.27 $\pm$ 0.50 mg CE/gDW for *R. officinalis* (Figure 1B). Thereby, tannins were less abundant than flavonoids in the leaves EEs of all species. The EE of *P. lentiscus*, *E. multiflora*, and *C. villosa* had the highest amounts of tannins, which are 17.36 $\pm$ 0.60, 25.82 $\pm$ 0.69, and 25.79 $\pm$ 3.01 of mg CE/gDW), respectively (Figure 1C).

### 3.2. Biological Activities

3.2.1. Determination of Antioxidant Activity

The antioxidant activity revealed significant differences ($p < 0.05$) between ethanolic extracts of the studied species. The total antioxidant capacity (TAC) was determined using the phosphomolybdenum method. The most important TAC was shown in *C. villosa* (15.33 $\pm$ 0.39 mg GAE/gDW) and *P. latifolia* (20.16 $\pm$ 0.32 mg GAE/gDW).

Indeed, in this part of the study, two tests were used: DPPH and FRAP. All the tests are based on electron or hydrogen transfer to assess the free radical scavenging activity. Results showed significant differences ($p < 0.05$) between ethanolic extracts (Table 1). Since $IC_{50}$ is in inverse proportion to the antioxidant capacity to antioxidant capacity, the lowest $IC_{50}$ value corresponds to the most potent antioxidant. Our results showed that the highest DPPH free radical reduction activity in EE species was detected for *C. villosa*, and the lowest were those corresponding to *P. lentiscus* and *E. multiflora*. Similarly, ferric reductase activity exhibited the same pattern compared to that of DPPH. *P. lentiscus* and *E. multiflora* showed the greatest ability to reduce ferric ions ($Fe^{3+}$).

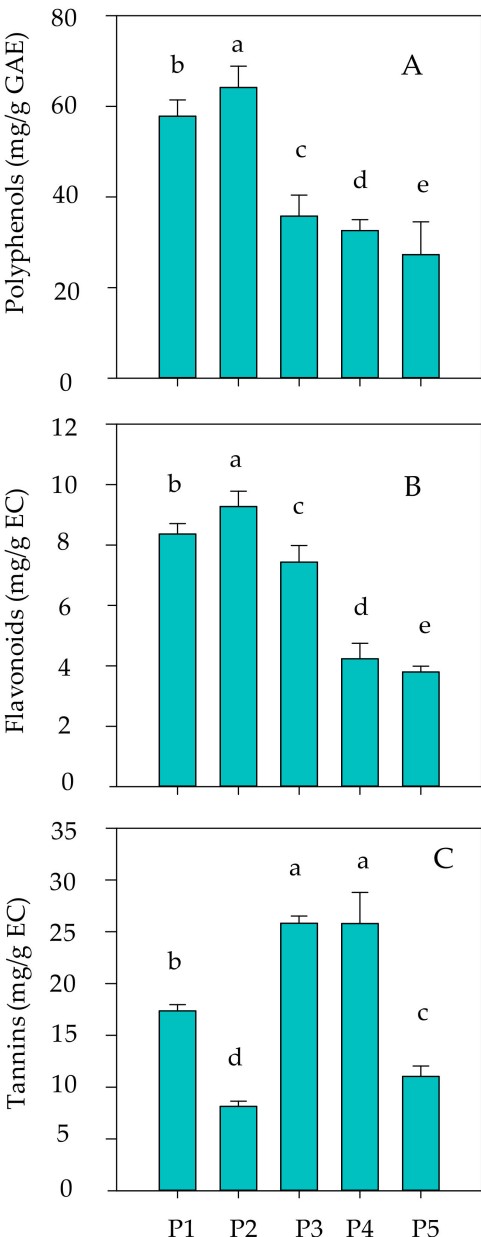

**Figure 1.** Total phenols (mg GAE/gDW; (**A**)), flavonoids (mg CE/gDW; (**B**)) and tannins (mg CE/gDW; (**C**)) content in P1 (*Pistacia lentiscus*), P2 (*Rosmarinus officinalis*), P3 (*Erica multiflora*), P4 (*Calicotome villosa*) and P5 (*Phillyrea latifolia*) ethanolic extracts. Means of three replicates are reported (±SD). Significant differences are detected in the bars ($p < 0.05$), Duncan's test. The letters (a–e) denote significant ($p < 0.05$) difference between means.

**Table 1.** Antioxidant activities of *Pistacia lentiscus*, *Rosmarinus officinalis*, *Erica multiflora*, *Calicotome villosa*, and *Phillyrea latifolia* ethanolic extracts.

|  | TAC (GAE/gDW) | DPPH (IC$_{50}$ mg/mL) | FRAP (EC$_{50}$ mg/mL) |
|---|---|---|---|
| *Pistacia lentiscus* | 13.50 [c] ± 1.50 | 5.38 [e] ± 0.44 | 13.39 [e] ± 1.29 |
| *Rosmarinus officinalis* | 12.10 [d] ± 0.53 | 12.25 [c] ± 0.24 | 20.82 [b] ± 0.52 |
| *Erica multiflora* | 13.45 [c] ± 2.48 | 10.85 [d] ± 0.99 | 17.89 [d] ± 0.66 |
| *Calicotome villosa* | 15.33 [b] ± 0.39 | 19.60 [a] ± 0.39 | 24.2 [a] ± 0.11 |
| *Phillyrea latifolia* | 20.16 [a] ± 0.32 | 15.59 [b] ± 0.66 | 19.35 [c] ± 0.26 |

TAC, DPPH, and FRAP values showed in this table were the mean of three replicates ± SD (n = 3). The values with different exponents in the same column (a–e) showed significant differences at $p < 0.05$.

### 3.2.2. Determination of Minimum Inhibitory and Bactericidal Concentrations

The results of antibacterial activity, evaluated by the presence or absence of inhibition zones, MIC, and MBC values, illustrated important significant differences ($p < 0.05$) between ethanolic extracts (Table 2). The obtained results indicated that the studied EEs showed significant antibacterial activity with inhibition zones and MICs varying from 13 to 16 mm and 0.04 to 25 mg/mL, respectively. EE of *P. lentiscus* was most effective against *Listeria monocytogenes* with a MIC of 0.04 mg/mL. Nevertheless, *E. multiflora* presented the same antibacterial potential against *S. aureus* and *E. coli*, and the MIC was 0.04 mg/mL (Table 2). Moreover, *Salmonella typhimurium* was almost inhibited by the same MIC of all EEs (3.84 mg/mL).

**Table 2.** Qualitative and quantitative results (MIC and BMC) of the antibacterial effects of the EEs of *Pistacia lentiscus*, *Rosmarinus officinalis*, *Erica multiflora*, *Calicotome villosa*, and *Phillyrea latifolia* on the reference strains were assayed.

| | Inhibition Zone Diameter (mm) | | MIC (mg/mL) | BMC (mg/mL) |
|---|---|---|---|---|
| | EE | Gentamicin (10 mg/disc) | | |
| *Staphylococcus aureus* | | | | |
| *Pistacia lentiscus* | 15 [c] ± 0.8 | | 12 | 25 |
| *Rosmarinus officinalis* | 14.5 ± 0.2 [cd] | | 3.84 | 12 |
| *Erica multiflora* | 16 [b] ± 0.1 | 24 [a] ± 0.1 | 0.04 | 3.84 |
| *Calicotome villosa* | 13 [d] ± 0.3 | | 25 | 25 |
| *Phillyrea latifolia* | 16 [b] ± 0.5 | | 3.84 | 12 |
| *Escherichia coli* | | | | |
| *Pistacia lentiscus* | 15 [c] ± 0.7 | | 12 | 25 |
| *Rosmarinus officinalis* | 14 [d] ± 0.00 | | 3.84 | 12 |
| *Erica multiflora* | 16 [b] ± 0.3 | 25 [a] ± 0.08 | 0.04 | 3.84 |
| *Calicotome villosa* | 15 [c] ± 0.5 | | 25 | 25 |
| *Phillyrea latifolia* | 13 [e] ± 0.5 | | 3.84 | 12 |
| *Listeria monocytogenes* | | | | |
| *Pistacia lentiscus* | 13 [d] ± 0.4 | | 0.04 | 3.84 |
| *Rosmarinus officinalis* | 14 [c] ± 0.5 | | 3.84 | 3.84 |
| *Erica multiflora* | 15 [b] ± 0.1 | 25 [a] ± 0.1 | 3.84 | 3.84 |
| *Calicotome villosa* | 14 [c] ± 0.3 | | 3.84 | 12 |
| *Phillyrea latifolia* | 15 [b] ± 0.5 | | 3.84 | 12 |
| *Salmonella typhimurium* | | | | |
| *Pistacia lentiscus* | 15 [b] ± 0.2 | | 3.84 | 12 |
| *Rosmarinus officinalis* | 13 [c] ± 0.4 | | 12 | 25 |
| *Erica multiflora* | 14 [c] ± 0.7 | 25 [a] ± 0.1 | 3.84 | 12 |
| *Calicotome villosa* | 14 [c] ± 0.2 | | 3.84 | 12 |
| *Phillyrea latifolia* | 14 [c] ± 0.5 | | 3.84 | 12 |

The values of EE, Gentamicin, MIC, and BMC showed in this table were the mean of three replicates ± SD (n = 3). The values with different exponents in the same row (a–e) showed significant differences at $p < 0.05$.

### 3.2.3. Mode of Action of Ethanolic Extract
Time Kill Assays

The speed and duration of antibacterial property of ethanolic extracts of studied plants against tested strains posted a significant difference ($p < 0.05$). The studied strains were evaluated by analysis of lysis and kill time. Bacterial counts were plotted against time to obtain the kill time graph (Figure 2). Data were presented in terms of colony-forming units. This kill kinetics study was performed against the studied strains at a concentration equivalent to the MIC and for 24 h against the strains studied at a concentration equivalent to the MIC.

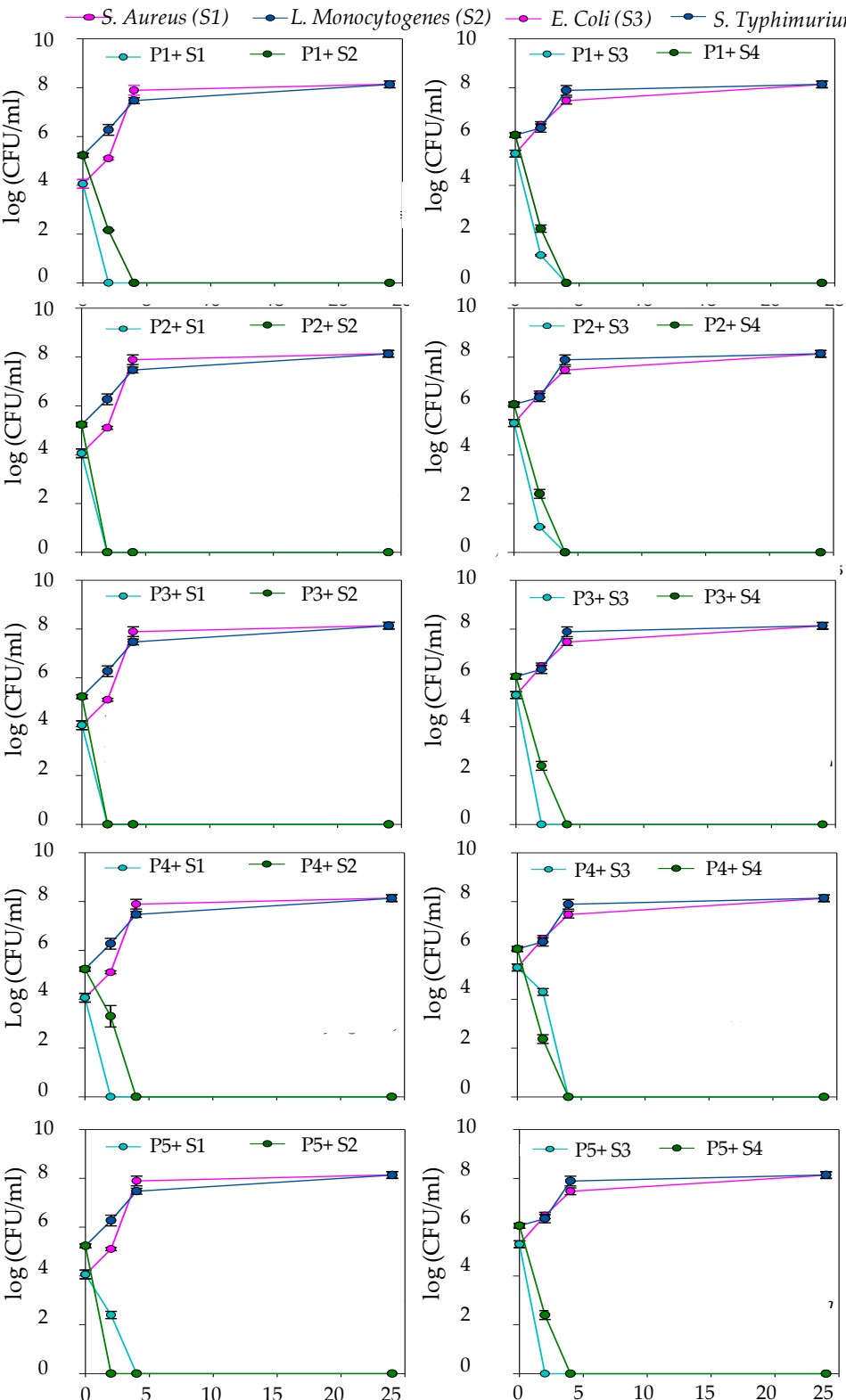

**Figure 2.** Time-kill curves of Gram-negative in the left (*Escherichia coli (S2)* and *Salmonella typhimurium (S4)*) and Gram-positive in the right (*Listeria monocytogenes (S2)* and *Staphylococcus aureus (S1)*) in control suspensions and the MICs of P1 *(Pistacia lentiscus)*, P2 *(Rosmarinus officinalis)*, P3 *(Erica multiflora)*, P4 *(Calicotome villosa)* and P5 *(Phillyrea latifolia)* ethanolic extracts. The mean of the standard deviations for three replicates is indicated by symbols. The lower detection threshold was 0 CFU/mL.

The control bacterial population exhibited a classic three-phase growth curve (growth, stationary, and decline). In the presence of each extract at a concentration corresponding to the MIC, the shape of the growth curve was inverted, and the three phases disappeared. This indicated the stop of growth of the four bacterial strains after 24 h incubation at 37 °C with the EEs at the MIC concentration.

The EE of *C. villosa*, *P. lentiscus*, *R. officinalis*, and *E. multiflora* caused the most significant decrease in the growth pattern of S. aureus within the first few minutes. Whereas the EEs of *P. latifolia* and *E. multiflora* produced a drop in the growth of *E. coli* at t = 2 h. Moreover, it was also observed that after exposure of the tested *L. monocytogenes* to the EE of *E. multiflora*, *P. latifolia*, and *R. officinalis*, the microorganisms showed lysis after 2 h.

Determination of the Lytic Action of Ethanolic Extracts

To determine the lytic action of *C. villosa*, *P. lentiscus*, *R. officinalis*, *E. multiflora*, and *P. latifolia* EEs on four bacterial species (*S. aureus*, *L. monocytogenes*, *S. Typhimurium*, and *E. coli*), the absorbance of the bacterial strains in the absence and presence of the extract at a concentration corresponding to MIC for 0, 60, 120 min, and 24 h were measured. The results revealed a significant difference ($p < 0.05$).

The obtained (Table 3) results indicated that in the case of the control (without extract), the absorbance of four bacterial strains was around 100%, signifying the absence of cell lysis. While the addition of extracts led to a diminution in the initial absorbance of all bacteria after 60 min of incubation. The EEs had lytic power against the strains. Indeed, optical density decreased after 2 h of incubation to ≈50 and 60% for all treated bacteria with EEs. Particularly, OD620s of treated suspensions were reduced to ≈25% with extracts of *E. multiflora*, *P. lentiscus*, and *P. latifolia* and to ≈35% with extracts of *C. villosa* and *R. officinalis*.

3.2.4. Loss of Salt Tolerance

The obtained results for the colony-forming capacity of E. coli, *L. monocytogenes*, *S. typhimurium*, and *S. aureus* in NA supplemented with 2.5% and 5% NaCl, respectively, after exposure to *C. villosa*, *E. multiflora*, *P. lentiscus*, *R. officinalis*, and *P. latifolia* EEs illustrated a significant difference ($p < 0.05$) (Table 4). Bacteria treated with ethanolic extracts of *R. officinalis* showed no bacterial growth in NA dishes supplemented with 2.5 and 5% NaCl. Indeed, *E. coli* treated with 5% NA and EEs of *P. lentiscus*, *P. latifolia*, and *E. multiflora* experienced a total growth arrest with cell death.

*3.3. Bioactive Compounds Investigation by HPLC Analysis*

In the literature, many compounds had been confirmed with reference standards in studied species using HPLC for its precision, versatility, and freedom from interferences [32]. According to HPLC data (Table 5), Turan and Mammadov [33] showed that vanillic acid, 4 hydroxybenzoic acid, and chlorogenic acid were the most important compounds in the extract of *the C. villosa* plant. Concerning *Rosmarinus officinalis*, previous research found that rosmarinic acid, carnosic acid, and chlorogenic acid were the principal compounds in the ethanolic extracts [12,34], while chlorogenic acid, 3,4,5 tri-*O*-galloyquinic acid, and rutin were the most abundant phenolic compounds in the ethanolic extract of *Pistacia lentiscus* in previous studies [10,11]. *Erica multiflora* was characterized by three major compounds, which were quercetin, rutin, and catechin [9]. The last plant was *Phillyrea latifolia*, which contained tyrosol, quercetin-7-*O*-rutinoside, and oleuropein [11].

**Table 3.** The proportion of initial OD620 of suspensions of *Escherichia coli*, *Salmonella typhimurium*, *Listeria monocytogenes*, and *Staphylococcus aureus* remaining after treatment with MICs of *Pistacia lentiscus*, *Rosmarinus officinalis*, *Erica multiflora*, *Calicotome villosa*, and *Phillyrea latifolia* ethanolic extracts for 60 min, 120 min, and 24 h.

| Treatment Agents (MICs) and Strains | OD 620 at (T) min/OD 620 at 0 min (%) | | | |
|---|---|---|---|---|
| | Immediately | After 60 min | After 120 min | After 24 h |
| Control: *S. aureus* | 103.1 [a] ± 6.17 | 94.7 [a] ± 4.28 | 91.7 [a] ± 1.31 | 102.5 [a] ± 4.23 |
| Control: *E. coli* | 98.2 [a] ± 0.66 | 99.1 [a] ± 0.04 | 95.6 [a] ± 1.70 | 98.4 [a] ± 0.78 |
| Control: *L. monocytogenes* | 98.22 [a] ± 0.66 | 99.21 [a] ± 0.8 | 98.35 [a] ± 0.90 | 99.40 [a] ± 0.41 |
| Control: *S. typhimurium* | 100.02 [a] ± 3.01 | 99.01 [a] ± 1.00 | 98.76 [a] ± 0.55 | 101.02 [a] ± 3.46 |
| *P. lentiscus* + *S. aureus* | 96.56 [a] ± 01.13 | 83.49 [b] ± 0.90 | 59.48 [c] ± 0.82 | 25.21 [d] ± 1.75 |
| *P. lentiscus* + *E. coli* | 98.43 [a] ± 0.51 | 67.61 [b] ± 0.90 | 51.03 [c] ± 1.05 | 27.61 [d] ± 1.78 |
| *P. lentiscus* + *L. monocytogenes* | 98.36 [a] ± 0.39 | 66.59 [b] ± 0.90 | 47.54 [c] ± 1.88 | 22.87 [d] ± 2.13 |
| *P. lentiscus* + *S. typhimurium* | 97.58 [a] ± 0.64 | 70.21[b] ± 0.33 | 56.31 [c] ± 1.04 | 22.22 [d] ± 1.57 |
| *R. offinalis* + *S. aureus* | 98.21 [a] ± 1.01 | 67.10 [b] ± 0.01 | 59.42 [bc] ± 0.74 | 31.14 [c] ± 1.22 |
| *R. offinalis* + *E. coli* | 98.77 [a] ± 0.68 | 80.37 [b] ± 0.85 | 49.17 [c] ± 1.50 | 35.4 [d] ± 0.75 |
| *R. offinalis* + *L. monocytogenes* | 98.43 [a] ± 0.74 | 80.74 [b] ± 0.70 | 50.52 [c] ± 0.08 | 25.6 [d] ± 1.37 |
| *R. offinalis* + *S. typhimurium* | 98.43 [a] ± 1.05 | 73.25 [b] ± 2.29 | 57.73 [c] ± 0.63 | 35.10 [d] ± 0.78 |
| *E. multiflora* + *S. aureus* | 97.49 [a] ± 0.14 | 66.77 [b] ± 1.67 | 57.98 [c] ± 2.22 | 31.53 [d] ± 2.33 |
| *E. multiflora* + *E. coli* | 97.96 [a] ± 0.50 | 67.48 [b] ± 0.22 | 53.54 [c] ± 2.87 | 25.48 [d] ± 3.56 |
| *E. multiflora* + *L. monocytogenes* | 97.63 [a] ± 0.31 | 56.22 [b] ± 1.49 | 55.55 [b] ± 1.03 | 23.81 [c] ± 0.54 |
| *E. multiflora* + *S. typhimurium* | 97.67 [a] ± 0.18 | 63.99 [b] ± 0.61 | 58.51 [b] ± 2.5 | 26.45 [c] ± 0.99 |
| *C. villosa* + *S. aureus* | 97.55 [a] ± 0.44 | 77.19 [b] ± 3.34 | 54.04 [c] ± 1.17 | 31.27 [d] ± 0.31 |
| *C. villosa* + *E. coli* | 98.82 [a] ± 0.91 | 67.32 [b] ± 1.89 | 52.46 [c] ± 4.34 | 30.26 [d] ± 0.94 |
| *C. villosa* + *L. monocytogenes* | 98.43 [a] ± 0.51 | 72.55 [b] ± 4.61 | 51.49 [c] ± 0.90 | 28.05 [d] ± 1.01 |
| *C. villosa* + *S. typhimurium* | 97.55 [a] ± 1.51 | 83.27 [b] ± 1.07 | 58.89 [c] ± 0.66 | 41.27 [d] ± 0.85 |
| *P. latifolia* + *S. aureus* | 96.41 [a] ± 0.72 | 82.70 [b] ± 5.17 | 58.67 [c] ± 0.48 | 26.00 [d] ± 2.16 |
| *P. latifolia* + *E. coli* | 98.00 [a] ± 0.39 | 61.00 [b] ± 6.27 | 54.20 [c] ± 2.50 | 24.92 [d] ± 5.73 |
| *P. latifolia* + *L. monocytogenes* | 97.63 [a] ± 0.31 | 65.09 [b] ± 0.31 | 56.69 [c] ± 5.63 | 27.75 [d] ± 0.48 |
| *P. latifolia* + *S. typhimurium* | 97.26 [a] ± 0.82 | 64.97 [b] ± 0.42 | 55.88 [c] ± 1.11 | 25.01 [d] ± 0.51 |

The values of OD620 at different times shown in this table were the mean of three replicates with mean standard ± SD (n = 3). Significant differences ($p < 0.05$) were indicated by the values with different exponents in the same line (a–d).

**Table 4.** Percentage of *Escherichia coli*, *Salmonella typhimurium*, *Listeria monocytogenes*, and *Staphylococcus aureus* cells that can develop colonies on NA, NA supplemented with 2.5% and 5% of NaCl after 30 min of treatment with MICs of *Pistacia lentiscus*, *Rosmarinus officinalis*, *Erica multiflora*, *Calicotome villosa*, and *Phillyrea latifolia* EEs.

| | Growth of Strains (%) | | | | | | | | | | | |
|---|---|---|---|---|---|---|---|---|---|---|---|---|
| | Control | | *P. lentiscus* | | *R. offinalis* | | *E. mutiflora* | | *C. villosa* | | *P. latifolia* | |
| Strains | 2.5 | 5 | 2.5 | 5 | 2.5 | 5 | 2.5 | 5 | 2.5 | 5 | 2.5 | 5 |
| *S. aureus* | 100 [a] | 100 [a] | 0.24 [b] | 0.06 [d] | 0.22 [b] | 0 | 0.23 [b] | 0 | 0.25 [b] | 0.14 [c] | 0.23 [b] | 0.18 [b] |
| *E. coli* | 100 [a] | 100 [a] | 0.11 [c] | 0 | 0.23 [b] | 0 | 0.23 [b] | 0 | 0.25 [b] | 0.1 [b] | 0.23 [b] | 0 |
| *L. monocytogenes* | 100 [a] | 100 [a] | 0.13 [c] | 0.03 [cd] | 0.17 [bc] | 0 | 0.14 [c] | 0.06 [b] | 0.13 [c] | 0.04 [c] | 0.22 [b] | 0.04 [c] |
| *S. typhimurium* | 100 [a] | 100 [a] | 0.22 [b] | 0.06 [d] | 0.10 [d] | 0.08 [d] | 0.13 [c] | 0.08 [d] | 0.25 [b] | 0.19 [b] | 0.19 [b] | 0.10 [c] |

The values of percentage growth of strains shown in this table were the mean of three replicates and were given as the mean ± SD (n = 3). Significant differences ($p < 0.05$) were indicated by the values with different exponents in the same line (a–d).

**Table 5.** Phenolics composition of ethanolic extract of studied species. Results are reported as the mean ± standard deviations (SD) of three measurements.

| Species | Major Compound | Concentrations (mg/g) | References |
|---|---|---|---|
| *Pistacia lentiscus* | Chlorogenic acid<br>3,4,5 Tri-O-galloyquinic acid<br>Rutin | 17.4 ± 1.9<br>15.9 ± 0.2<br>13.6 ± 1.5 | [10,11] |
| *Phillyrea latifolia* | Tyrosol<br>Quercetin-7-O-rutinoside<br>Oleuropein | 78.2 ± 18.4<br>42.5 ± 5.1<br>167.0 ± 7.7 | [11] |
| *Erica multiflora* | Quercetin<br>Rutin<br>Catechin | 43.57 ± 0.18<br>42.05 ± 2.67<br>27.29 ± 2.73 | [9] |
| *Rosmarinus officinalis* | Rosmarinic acid<br>Carnosic acid<br>Chlorogenic acid | 38.5 ± 0.04<br>26.4 ± 0.15<br>11.2 ± 0.17 | [12,34] |
| *Calicotome villosa* | vanillic acid<br>4-hydroxybenzoic acid<br>chlorogenic acid | 1.4<br>0.39<br>0.39 | [33] |

## 4. Discussion

In this study, the secondary metabolites content (including total polyphenol, flavonoid, and tannin) of the EEs of five Tunisian plants—*Pistacia lentiscus*, *Rosmarinus officinalis*, *Erica multiflora*, *Calicotome villosa*, and *Phillyrea latifolia* were measured. Moreover, the antioxidant and antimicrobial activities of these five EEs were evaluated, and finally, the mechanisms of action were studied.

The studied plants were a great reservoir of phenolic compounds (Table 5). They were among the most versatile herbs that have been studied for their pharmacological and therapeutic effects. Tested plants were characterized by their high content of total polyphenols, tannins, and flavonoids [35]. The highest values of these biomolecules were demonstrated in *P. lentiscus*, followed by *R. officinalis*. However, *E. multiflora* and *C. villosa* showed the highest values of tannins. Similarly, previous research had found a similar richness in phenolic compounds [36–38]. Phenolic compounds are important biomolecules in plants due to their capacity to eliminate free radicals [39]. The most important total activity antioxidant was shown higher in *C. villosa* (15.33 ± 0.39 mg GAE/gDW) and *P. latifolia* (20.16 ± 0.32 mg GAE/gDW). Additionally, the antioxidant activities of the EEs of the studied plants were evaluated by two different assay methods, which were DPPH and FRAP assays. The EEs of *P. lentiscus* and *E. multiflora* showed the highest DPPH inhibition capacity. Similarly, the latter plants showed the highest iron-reducing capacity. These results were in accordance with previous data indicating a pronounced antioxidant effect of the studied plants [40]. This potentiality is mainly correlated with their richness in phenolic compounds known for their important reactivity as hydrogen or electron donors [41–44].

The antibacterial activity of EEs from the leaves of *Rosmarinus officinalis*, *Erica multiflora*, *Pistacia lentiscus*, *Calicotome villosa*, and *Phillyrea latifolia* was determined by measuring the zone of inhibition made against important pathogens (*Escherichia coli*, *Salmonella typhimurium*, *Listeria monocytogenes* and *Staphylococcus aureus*). The intensity of antibacterial activity was estimated by the double serial dilution method to determine the minimum inhibitory concentration (MIC) of the ethanolic extracts, whereas the bactericidal activity was determined by the in vitro killing time test, salt tolerance, and bacteriolysis.

The results demonstrated that EEs had good antibacterial activity at different concentrations against Gram-positive and Gram-negative bacteria (Table 2). The agar diffusion assay revealed that the bacteria were inhibited by the ethanolic extract to produce concentration-dependent zones of inhibition. At the MIC utilized, all bacteria strains were inhibited. The zones of inhibition produced by 0.04 to 3.84 mg/mL of EEs ranged from

13 mm to 16 mm for all bacteria. Among the plants that had a high degree of antibacterial activity (MIC = 0.04 mg/mL), the most important were *P. lentiscus* and *E. multiflora*. Concerning BMC, the values were the same or 2 to 4 times higher than the similar or 2 to 4 times higher than the MIC values of EEs. The variation between the activities of the EEs and the standard antimicrobial medicament may be attributed to the combinations of bioactive compounds that exist in the extract as compared to the pure compounds present in the standard antibiotics, which rarely have the same potency as the unrefined extract at comparable concentrations or doses of the active substance [45]. Previous studies had shown that Gram-positive bacterial strains were known to be more sensitive to plant extracts than Gram-negative bacterivores, which are more resistant [46–48]. This sensitivity is attributed to the availability of hydrophobic lipopolysaccharides in the outer membrane of Gram-negative bacteria. Thus, Gram-negative bacteria are more difficult to control due to their double-layered membrane [49]. Furthermore, our results highlight that the EEs were quite effective in controlling the bacteria compared to other previous studies [42,50]. None of our EEs demonstrated selective antimicrobial activity based on differences in bacterial cell walls.

Antimicrobial compounds in EEs of plants can inhibit bacterial growth through different mechanisms. Interactions with hydrophobic structures of bacterial strains had a key role in the antimicrobial actions of secondary plant metabolites [51,52]. In the present study, *L. monocytogenes* and *E. coli* cells were destroyed and killed in the stationary phase of growth (after 2 h) by *P. latifolia*, *R. officinalis*, and *E. multiflora* EEs, while *C. villosa*, *E. multiflora*, and *P. lentiscus* EEs inhibited the growth of *S. aureus* after 2 h. *S. typhimurium* was the only bacterium inhibited by all EEs after 4 h. In this growth phase, organisms are generally less susceptible to injury than those in the exponential phase [52]. Antimicrobial agents affecting cell synthesis processes often have little impact on organisms in the stationary phase of growth [53], and these results suggest that that the macromolecular synthesis process was not the primary target of EEs.

In recent years, several studies have attempted to identify and explain the mechanisms of action of certain medicinal plant extracts on pathogenic bacterial strains. The essential oils of rosewood, oregano, thyme [54], and tea tree [31] have been described as producing damage to the bacterial membrane and inducing cell lysis. Additionally, the essential oil of *B. pandurata* has been signaled to affect the membrane permeability of *Escherichia coli* [55]. In the present study, it was demonstrated that treatment of the studied bacterial cells with the studied plant EEs induced cell lysis. This possible mechanism of action may be related to the cell wall and membrane. Certain antimicrobial substances produce significant damage to the membrane and lead to lysis of the whole cell [31]. This phenomenon has been described previously for *Jatropha curcas*, oregano, and thyme EEs [54]. The inability of *Pistacia lentiscus*, *Rosmarinus officinalis*, *Erica multiflora*, *Calicotome villosa*, and *Phillyrea latifolia* or their components to lyse *Escherichia coli*, *Salmonella typhimurium*, *Listeria monocytogenes*, and *Staphylococcus aureus* cells suggested that cell wall destruction is not their principal mechanism of action. Treatment-induced release of autolytic enzymes from the membrane-bound cell wall eventually induces lysis [31], which may explain the delayed lysis of *Escherichia coli*, *Salmonella typhimurium*, *Listeria monocytogenes*, and *Staphylococcus aureus* noticed when suspensions were re-examined after several hours. In the same context, previous research had revealed that flavonoids could inhibit pathogen spore germination [56]; tannins are therefore likely to inhibit extracellular microbial enzymes, to derive the substrates necessary for microbial proliferation or to have a direct action on microbial metabolism by inhibiting oxidative phosphorylation [57]. The antimicrobial activity exhibited in our study could be due to the abundant content of phenolic compounds such as flavonoids and condensed tannins, known as antibacterial agents, in the EEs of the plants studied. In addition, the antibacterial activity of the studied plant extracts could be due to the existence of their major compounds (Table 5) or synergy between these compounds against various species of bacteria studied. Synergistic effects can be created if constituents of an extract affect different targets or have an interaction with each other to

enhance the solubility and, thereby, the bioavailability of one or more constituents of an extract [58].

The results of the salt tolerance test showed that the strains could tolerate a high concentration of NaCl (5%). *Escherichia coli*, *Salmonella typhimurium*, *Listeria monocytogenes* and *Staphylococcus aureus* are halophilic bacteria tolerating hyperosmotic conditions by increasing solutes inside their cells such as betaine, glycine, ectoine, glutamate, proline, and trehalose [59]. The outer membrane determines the capacity of bacteria to have the ability to tolerate a particular change in a set of ionic media [60]. Tolerance to salts or other potentially toxic compounds induces membrane damage or weakening and leads to cell lysis by structural disruption, increased cell leakage [61,62], and loss of osmoregulation of bacterial cells to salts [63]. Treatment of strains studied with *C. villosa* and *P. latifolia* and their components substantially decreased the capacity of survivors to form colonies on media with NaCl. While treatment of strains with *R. officinalis* and its components completely inhibited the ability to develop colonies on NaCl-containing media.

The obtained results from this investigation suggested that the studied plant ethanolic extracts could be useful as naturally occurring bioactive compounds in food preservation and thus in order to inhibit or monitor the growth of spoilage and pathogenic microorganisms. In addition, we will continue on the same axis of research using clinical and veterinary strains.

## 5. Conclusions

*P. lentiscus*, *R. officinalis*, *E. multiflora*, *C. villosa*, and *P. latifolia* were considered important medicinal plants and were used to treat studied bacterial strains and to highlight the modes of action of their ethanolic extracts. Gathered results clearly showed that *P. lentiscus* and *R. officinalis* were the richest species in flavonoids and phenolic compounds. While the EEs of *P. lentiscus*, *E. multiflora*, and *C. villosa* had the greatest amounts of tannins. In addition, *P. lentiscus* and *E. multiflora* showed the highest FRAP and DPPH potentials. Then, the higher TAC was registered in *Phillyrea latifolia* and *Calicotome villosa*.

The studied EEs presented important antibacterial activities against *Escherichia coli*, *Salmonella typhimurium*, *Listeria monocytogenes* and *Staphylococcus aureus* strains. Among the plants that had a high degree of antibacterial activity were *P. lentiscus* and *E. multiflora*. From the kinetics of time-kill tests, we can conclude that our EEs are very efficient at inhibiting and killing bacterial strains with an average duration of 2 or 4 h. In addition, EEs have lytic power against strains. Bacteria organisms treated with EEs at MICs showed a loss of tolerance to salt. This has encouraged their utilization in alternative medicine for the treatment of infections and their potential application as antimicrobial and antioxidant agents. The antibacterial properties and capacity of these plant extracts can be exploited to treat infections as an alternative to conventional antibiotics.

**Author Contributions:** Methodology, M.B.J.; Project administration, M.B.; Supervision, K.M. and Z.N.; Writing—original draft, K.N.; Writing—review & editing, S.D. All authors have read and agreed to the published version of the manuscript.

**Funding:** This research received no external funding.

**Informed Consent Statement:** Not applicable.

**Conflicts of Interest:** The authors declare no conflict of interest.

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
