# Peer review of "In Vitro Antioxidant, Antibacterial and Mechanisms of Action of Ethanolic Extracts of Five Tunisian Plants against Bacteria"

_applsci, doi:10.3390/app12105038_

Round 1

Reviewer 1 Report

This work provides a report on various biological activities of 5 plant extracts. While not novel in terms of methods or approach it does provide additional information about Tunisian plant species.

There are a number of areas for improvement:

  1. There are many grammatical errors and at times this hinders understanding of the work.
  2. Please ensure all abbreviations are given in full the first time used
  3. In the abstract please provide statistical information to support claims of higher/lower activity
  4. Introduction - this section is quite disjointed and does not present a cohesive narrative regarding the background literature and rationale for this study 
  5. Section 2.1 - it is unclear whether the plants were obtained from the field or an existing collection. How were the plants verified?
  6. What is the rationale for the selection of the specific organisms used in this study?
  7. What is the rationale for 3 replicates, I appreciate that this is commonly used however what is the statistical basis for this experimental design decision (ie is 3 sufficient to ensure statistical validity as determined by a  sample size calculation)
  8. Please provide exact p values (except for p<0.001)
  9. Fig 1 - it is unclear what the letters on the plot mean
  10. Table 2 - why is there no SD for R. officinalis and gentamicin? What is the 'n' for the data in this table
  11. Figure 2 - the figures are quite small and difficult to read, further it is unclear what each of the plots represents. Please give exact 'n' not 'at least three replicates'
  12. Please provide statistical analysis for the data in Table 2 
  13. In the tables where legends state that values with different superscripts are significantly different it is unclear what the actual comparisons are, are they row or column or both?
  14. What do the superscripts in Table 4 mean? if they are statistical significance then please provide SD and n values for all data 
  15. While the Conclusions are broadly supported by the data it is difficult for readers to judge the detail due to the issues with the presentation of the ressults.

Author Response

Response to Reviewer 1

I am very glad to receive your suggestions and recommendations for my paper and I hope that my manuscript will be published in your honorable Journal “Applied Sciences”. I check my paper point by point according to your suggestions and recommendations, and I answered to Reviewer’s comments. It’s noticed that all the modification and changes in the text body, tables and figures are highlighted in yellow colour.

Reviewer 1 suggest that « Extensive editing of English language and style required »

In response, I have adress my manuscript to a colleague native English speaking that she correct many grammatical errors and spelling. Please see manuscript.

Comments and Suggestions for Authors

This work provides a report on various biological activities of 5 plant extracts. While not novel in terms of methods or approach it does provide additional information about Tunisian plant species.

There are a number of areas for improvement:

  1. There are many grammatical errors and at times this hinders understanding of the work.

Response : we correct all of them, please see manuscript

  1. Please ensure all abbreviations are given in full the first time used
  1. Response : we correct all of them, please see manuscript
  1. In the abstract please provide statistical information to support claims of higher/lower activity
  1. Response : we correct all of them, please see manuscript
  1. Introduction - this section is quite disjointed and does not present a cohesive narrative regarding the background literature and rationale for this study 
  1. Response : we have insert many modifications and insertions in the introduction section, please see manuscript
  1. Section 2.1 - it is unclear whether the plants were obtained from the field or an existing collection. How were the plants verified?

Response : as indicated in the plant material section te leaves of the 5 plants were collected during spring (March 27th) of 2021 from the region of Zaghouan (northeast of Tunisia: 36.4281181110241N, 10.093329064007772E),, please see manuscript

  1. What is the rationale for the selection of the specific organisms used in this study?

Response : Because the selected microorganisms are the main representant of the Gram positive and the gram negative bacteria. Its noticed that some pathogens like these bacteria have multidrug resistance to the currently used antibiotics triggered the search for identifying natural antimicrobial products that are effective in combating infections

  1. What is the rationale for 3 replicates, I appreciate that this is commonly used however what is the statistical basis for this experimental design decision (ie is 3 sufficient to ensure statistical validity as determined by a  sample size calculation)

Response : we use 3 replications that we think that there are enough for statistical treatment to determine standard deviation (SD) and ANOVA folowed by Duncan’s test in order to determine the significance between means. So, we think that simple size is aduequate to validate obtained statistical results.

  1. Please provide exact p values (except for p<0.001)

Response : it’s a mistake and I correct it to become P < 0.05.

  1. Fig 1 - it is unclear what the letters on the plot mean

Response : yes, I correct Figure 1 letters of significance to be more clear and I add in the Figure 1 footnote (The letters (a-e) denote significant (P < 0.05) difference between means).

  1. Table 2 - why is there no SD for R. officinalis and gentamicin? What is the 'n' for the data in this table

Response : sorry it’s a mistake, I have calculate the SD for R. ofiicinalis and for gentamicin and we add them, please see Table 2. The n for the data in this table was 3.

  1. Figure 2 - the figures are quite small and difficult to read, further it is unclear what each of the plots represents. Please give exact 'n' not 'at least three replicates'

Response : I modify Figure 2 in order to become more clear and readable.

Axis x express time of bacteria killig et the y one express the log cfu/mL. The exact n was 3 : n=3.

  1. Please provide statistical analysis for the data in Table 2 

Response : it’s ok, please see Table 2.

  1. In the tables where legends state that values with different superscripts are significantly different it is unclear what the actual comparisons are, are they row or column or both?

Response : for Table 1 and Table the comparaison are in the row, but in Table 3 and Table 4, the comparaisons were in the lines.

  1. What do the superscripts in Table 4 mean? if they are statistical significance then please provide SD and n values for all data 

Response : it’s ok, please see Table 4.

  1. While the Conclusions are broadly supported by the data it is difficult for readers to judge the detail due to the issues with the presentation of the ressults.

Response : it’s ok, please see conclusion section.

With my best regards

Prof. Kamel Msaada

Reviewer 2 Report

Dear authors,

The manuscript entitled "In Vitro Antioxidant, Antibacterial and Mechanisms of Action of Ethanolic Extracts of Five Tunisian Plants Against Bacteria" was reviewed. It details the experimental procedure for the extraction of antioxidant and antibacterial compounds from plants native to Tunisia. The work is attractive and well structured, finding very few spelling and formatting errors (highlighted in the revised PDF file of the manuscript). The discussion is solid and interesting. I suggest that the part of the introduction may be enriched, especially the justification, by including some previously reported works. For this purpose, please refer to the following publications that I consider should be reviewed and included in the corrected version of your manuscript.

https://doi.org/10.1016/j.indcrop.2012.04.011

https://doi.org/10.3389/fmicb.2018.01639

https://doi.org/10.5530/pj.2018.1.28

https://doi.org/10.3390/molecules26071973

https://doi.org/10.1002/jobm.200410534

https://doi.org/10.5897/AJMR.9000402

https://doi.org/10.3389/fmicb.2015.00577

Best regards

Author Response

Response to Reviewer 2

I am very glad to receive your suggestions and recommendations for my paper and I hope that my manuscript will be published in your honorable Journal “Applied Sciences”. I check my paper point by point according to your suggestions and recommendations, and I answered to Reviewer’s comments. It’s noticed that all the modification and changes in the text body, tables and figures are highlighted in yellow colour.

Comments and Suggestions for Authors

Dear authors,

The manuscript entitled "In Vitro Antioxidant, Antibacterial and Mechanisms of Action of Ethanolic Extracts of Five Tunisian Plants Against Bacteria" was reviewed. It details the experimental procedure for the extraction of antioxidant and antibacterial compounds from plants native to Tunisia. The work is attractive and well structured, finding very few spelling and formatting errors (highlighted in the revised PDF file of the manuscript). The discussion is solid and interesting. I suggest that the part of the introduction may be enriched, especially the justification, by including some previously reported works. For this purpose, please refer to the following publications that I consider should be reviewed and included in the corrected version of your manuscript.

https://doi.org/10.1016/j.indcrop.2012.04.011

https://doi.org/10.3389/fmicb.2018.01639

https://doi.org/10.5530/pj.2018.1.28

https://doi.org/10.3390/molecules26071973

https://doi.org/10.1002/jobm.200410534

https://doi.org/10.5897/AJMR.9000402

https://doi.org/10.3389/fmicb.2015.00577

I have add all of these references in my manuscript in the appropriate place.

Comments from pdf file :

C1-Please reduce the length of the Abstract to lees than 250 words.

Response : its ok, abstract word count was reduced to be under 250 words

C2-spaces

Response : I revise all the missing spaces in the whole text body. Please see manuscript

C-3 In order to improve the justification, after this section you may add the articles suggested.

Response : its ok, please see manuscript

C-4 Please check the format of the text in the results section, since it seems to be shifted to the right relative to section 2.6

Response : its ok, please see manuscript

C-5 Please try to rearrange the symbols in these figures. They are hard to follow in the electronic version of the manuscript, but almost impossible in the printed version.

Response : its ok, please see Figure 2

C-6 Please improve the conclusions. These conclusions seems more as a summary of the results.

Response : its ok, please see conclusion

Best regards

Prof. Kamel Msaada

Reviewer 3 Report

To the authors, I have some observations:
- I ask the authors to check the configuration of the article because some words are without spaces (example, line 15);
- check the references used, the authors start the introduction with a comment as a reference (2006), I suggest updating and using revision works, which are more suitable;
- introduction - bring more information about the analyzes carried out, in a succinct way, in order to better justify the study and the activities evaluated (Antioxidant, Antibacterial..)
- 9 keywords seem excessive to me, I suggest  reducing and being more punctual;
- insert a closing sentence in the abstract;
- aerial part (line 62) consists of leaves, flowers, inflorescences, twigs? explain, please;
- indicate the geographic location of where the collection took place;
- indicate the p-value in the statistical analysis and describe the use of Duncan's test;
- review the spelling of Latin words, configure (example, title of table 1)
- I suggest that the authors indicate the limitations of the study at the end of the discussion.

Author Response

Response to Reviewer 3

I am very glad to receive your suggestions and recommendations for my paper and I hope that my manuscript will be published in your honorable Journal “Applied Sciences”. I check my paper point by point according to your suggestions and recommendations, and I answered to Reviewer’s comments. It’s noticed that all the modification and changes in the text body, tables and figures are highlighted in yellow colour.

Comments and Suggestions for Authors

To the authors, I have some observations:

- I ask the authors to check the configuration of the article because some words are without spaces (example, line 15);

Response : I check my manuscript point by point and I correct all of them accordingly.

- check the references used, the authors start the introduction with a comment as a reference (2006), I suggest updating and using revision works, which are more suitable;

Response : yes I upadate my manuscript with regard to the used references and I revise all the manuscript.

- introduction - bring more information about the analyzes carried out, in a succinct way, in order to better justify the study and the activities evaluated (Antioxidant, Antibacterial..).

Response : yes I upadate my manuscript with regard to the used references and I revise all the manuscript.

- 9 keywords seem excessive to me, I suggest  reducing and being more punctual;

Response : It’s ok we reduce them to become : Medicinal plants, Ethanolic extracts, Antioxidant activities, antibacterial activity, antibacterial mechanism of actions.

- insert a closing sentence in the abstract.

Response : it’s ok, please see abstract.

- aerial part (line 62) consists of leaves, flowers, inflorescences, twigs? explain, please;

Response : Yes you are right, we use only leaves, we correct all of them in the whole manuscript.

- indicate the geographic location of where the collection took place;

Response :it’s ok, please see manuscript (Plant Material section).

- indicate the p-value in the statistical analysis and describe the use of Duncan's test;

Response :it’s ok, please see manuscript (statistical analysis section).

- review the spelling of Latin words, configure (example, title of table 1)

Response :it’s ok, please see manuscript

- I suggest that the authors indicate the limitations of the study at the end of the discussion.

Response :it’s ok, please see manuscript (the end of the discussion section)

With my best regards

Prof. Kamel Msaada
